# TRANSIENT NON-STATIONARITY AND GENERALISATION IN DEEP REINFORCEMENT LEARNING

**Maximilian Igl** [*]  **Gregory Farquhar** [*,†]  **Jelena Luketina** [*]  **Wendelin Böhmer** [‡]

**Shimon Whiteson** [*]

## ABSTRACT

Non-stationarity can arise in Reinforcement Learning (RL) even in stationary environments. For example, most RL algorithms collect new data throughout training, using a non-stationary behaviour policy. Due to the transience of this non-stationarity, it is often not explicitly addressed in deep RL and a single neural network is continually updated. However, we find evidence that neural networks exhibit a memory effect where these transient non-stationarities can permanently impact the latent representation and adversely affect generalisation performance. Consequently, to improve generalisation of deep RL agents, we propose Iterated Relearning (ITER). ITER augments standard RL training by repeated knowledge transfer of the current policy into a freshly initialised network, which thereby experiences less non-stationarity during training. Experimentally, we show that ITER improves performance on the challenging generalisation benchmarks *ProcGen* and *Multiroom*.

## 1 INTRODUCTION

In RL, as an agent explores more of its environment and updates its policy and value function, the data distribution it uses for training changes. In deep RL, this non-stationarity is often not addressed explicitly. Typically, a single neural network model is initialised and continually updated during training. Conventional wisdom about catastrophic forgetting (Kemker et al., 2018) implies that old updates from a different data-distribution will simply be forgotten. However, we provide evidence for an alternative hypothesis: networks exhibit a memory effect in their learned representations which can harm generalisation permanently if the data-distribution changed over the course of training.

To build intuition, we first study this phenomenon in a supervised setting on the CIFAR-10 dataset. We artificially introduce transient non-stationarity into the training data and investigate how this affects the asymptotic performance under the final, stationary data in the later epochs of training. Interestingly, we find that while asymptotic training performance is nearly unaffected, test performance degrades considerably, even after the data-distribution has converged. In other words, we find that latent representations in deep networks learned under certain types of non-stationary data can be inadequate for good generalisation and might not be improved by later training on stationary data.

Such transient non-stationarity is typical in RL. Consequently, we argue that this observed degradation of generalisation might contribute to the inferior generalisation properties recently attributed to many RL agents evaluated on held out test environments (Zhang et al., 2018a;b; Zhao et al., 2019). Furthermore, in contrast to Supervised Learning (SL), simply re-training the agent from scratch once the data-distribution has changed is infeasible in RL as current state of the art algorithms require data close to the on-policy distribution, even for off-policy algorithms like Q-learning (Fedus et al., 2020).

To improve generalisation of RL agents despite this restriction, we propose Iterated Relearning (ITER). In this paradigm for deep RL training, the agent's policy and value are periodically distilled into a freshly initialised student, which subsequently replaces the teacher for further optimisation. While this occasional distillation step simply aims to re-learn and replace the current policy and

---

[*]University of Oxford. Corresponding author: Maximilian Igl (maximilian.igl@gmail.com)

[†]Now at DeepMind, London

[‡]Delft University of Technology

value outputs for the training data, it allows the student to learn a better latent representation with improved performance for unseen inputs because it eliminates non-stationarity during distillation. We propose a practical implementation of ITER which performs the distillation in parallel to the training process without requiring additional training data. While this introduces a small amount of non-stationarity into the distillation step, it greatly improves sample efficiency without noticeably impacting performance.

Experimentally, we evaluate ITER on the *Multiroom* environment, as well as several environments from the recently proposed *ProcGen* benchmark and find that it improves generalisation. This provides further support to our hypothesis and indicates that the non-stationarity inherent to many RL algorithms, even when training on stationary environments, should not be ignored when aiming to learn robust agents. Lastly, to further support this claim and provide more insight into possible causes of the discovered effect, we perform additional ablation studies on the CIFAR-10 dataset.

## 2 BACKGROUND

We describe an RL problem as a Markov decision process (MDP) $(\mathcal{S}, \mathcal{A}, T, r, p_0, \gamma)$ (Puterman, 2014) with actions $a \in \mathcal{A}$, states $s \in \mathcal{S}$, initial state $s_0 \sim p_0$, transition dynamics $s' \sim T(s, a)$, reward function $r(s, a) \in \mathbb{R}$ and discount factor $\gamma$. The unnormalised discounted state distribution induced by a policy $\pi$ is defined as $d_\pi(s) = \sum_{t=0}^{\infty} \gamma^t \mathrm{Pr}\left(S_t = s | S_0 \sim p_0, A_t \sim \pi(\cdot | S_t), S_{t+1} \sim T(S_t, A_t)\right)$. In ITER, we learn a sequence of policies and value functions, which we denote with $\pi^{(k)}(a|s)$ and $V^{(k)}(s)$ at the $k$th iteration ($k \in \{0, 1, 2, \dots\}$), parameterized by $\theta_k$.

We briefly discuss some forms of non-stationarity which can arise in RL, even when the environment is stationary. For simplicity, we focus the exposition on actor-critic methods which use samples from interaction with the environment to estimate the policy gradient given by $g = \mathbb{E}[\nabla_\theta \log \pi_\theta(a|s) A^\pi(s, a, s') | s, a, s' \sim d_\pi(s)\pi(a|s)T(s'|s, a)]$. The advantage is often estimated as $A^\pi(s, a, s') = r(s, a) + \gamma V^\pi(s') - V^\pi(s)$. Typically, we also use neural networks to approximate the baseline $V_\phi^\pi(s)$ and for bootstrapping from the future value $V_\phi^\pi(s')$. $\phi$ can be learned by minimising $\mathbb{E}[A^\pi(s, a, s')^2]$ by stochastic semi-gradient descent, treating $V_\phi^\pi(s')$ as a constant.

There are at least three main types of non-stationarity in deep RL. First, we update the policy $\pi_\theta$, which leads to changes in the state distribution $d_{\pi_\theta}(s)$. Early on in training, a random policy $\pi_\theta$ only explores states close to initial states $s_0$. As $\pi_\theta$ improves, new states further from $s_0$ are encountered. Second, changes to the policy also change the true value function $V^\pi(s)$ which $V_\phi^\pi(s)$ is approximating. Lastly, due to the use of bootstrap targets in temporal difference learning, the learned value $V_\phi^\pi(s)$ is not regressed directly towards $V^\pi(s)$. Instead $V_\phi^\pi$ fits a gradually evolving target sequence even under a fixed policy $\pi$, thereby also changing the policy gradient estimator $g$.

## 3 THE IMPACT OF NON-STATIONARITY ON GENERALISATION

In this section we investigate how asymptotic performance is affected by changes to the data-distribution during training. In particular, we assume an initial, transient phase of non-stationarity, followed by an extended phase of training on a stationary data-distribution. This is similar to the situation in RL where the data-distribution is affected by a policy which converges over time. We show that this transient non-stationarity has a permanent effect on the learned representation and negatively impacts generalisation. As interventions in RL training can lead to confounding factors due to off-policy data or changed exploration behaviour, we utilise Supervised Learning (SL) here to provide initial evidence in a more controlled setup. We use the CIFAR-10 dataset for image classification (Krizhevsky et al., 2009) and artificially inject non-stationarity.

Our goal is to provide qualitative results on the impact of non-stationarity, not to obtain optimal performance. We use a ResNet18 (He et al., 2016) architecture, similar to those used by Espeholt et al. (2018) and Cobbe et al. (2019a). Parameters are updated using Stochastic Gradient Descent (SGD) with momentum and, following standard practice in RL, we use a constant learning rate and do not use batch normalisation. Weight decay is used for regularisation. Hyper-parameters and more details can be found in appendix B.

We train for a total of 2500 epochs. While the last 1500 epochs are trained on the full, unaltered dataset, we modify the training data in three different ways during the first 1000 epochs. Test data is

left unmodified throughout training. While each modification is loosely motivated by the RL setting, our goal is not to mimic it exactly (which would be infeasible), nor to disentangle the contributions of different types of non-stationarity. Instead, we aim to show that these effects reliably occur in the presence to various types of non-stationarity, and provide intuitions that can be brought into the RL setting in Section 4.

For the first modification, called `Dataset Size`, we initially train only on a small fraction of the full dataset and gradually add more data points after each epoch, at a constant rate, until the full dataset is available after epoch 1000. During the non-stationary phase, data points are reused multiple times to ensure the same number of network updates are made in each epoch. For the modification `Wrong Labels` we replace all training labels by randomly drawn incorrect ones. After each epoch, a constant number of these labels are replaced by their correct values. Lastly, `Noisy Labels` is similar to `Wrong Labels`, but the incorrect labels are sampled uniformly at the start of each epoch. For both, all training labels are correct after epoch 1000. While `Dataset Size` is inspired by the changing state distribution seen by an evolving policy, `Wrong Labels` and `Noisy Labels` are motivated by the consistent bias or fluctuating errors a learned critic can introduce in the policy gradient estimate.

The results are shown in fig. 1. While the final training accuracy (left) is almost unaffected (see table 1 in the appendix for exact results), all three non-stationary modifications significantly reduce the test accuracy (right). The plateau in accuracy shows that this effect persists even after the models are further trained using the full dataset with correct labels: non-stationarity early in training has a permanent effect on the learned representations and quality of generalisation. These results indicate that the non-stationarity introduced by the gradual convergence of the policy in RL might similarly de-

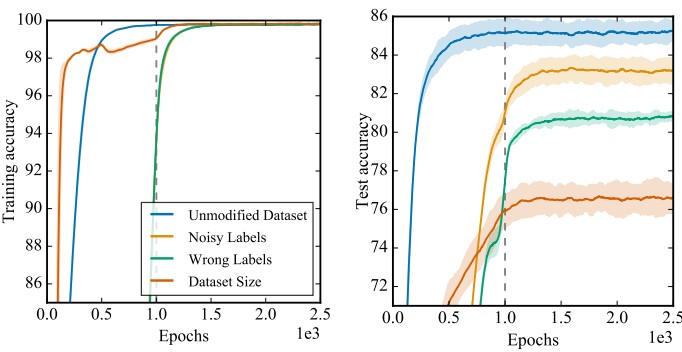

**Figure 1:** Accuracy on CIFAR-10 when the training data is non-stationary over the first 1000 epochs (dashed line). The remaining epochs are trained on the full, unaltered training data. Testing is performed on unaltered data throughout. While final training performance (left) is almost unaffected, test accuracy (right) is significantly reduced by initial, transient non-stationarity.

teriorate the generalisation of the agent. To overcome this, we propose ITER in the next section. The key insight enabling ITER is that the observed negative effect is restricted to the test data, whereas the predictions on the training data are unaffected and of high quality.

## 4    ITER

In section 3, we have seen evidence that the non-stationarity which is present in many deep RL algorithms might lead to impaired generalisation on held-out test environments. To mitigate this and improve generalisation to previously unseen states, we propose Iterated Relearning (ITER): instead of updating a single agent model throughout the entire training process, ITER learns a sequence of models, each of which is exposed to less non-stationarity during its training. As we will show in section 5, this improves generalisation. ITER can be applied on top of a wide range of base RL training methods. For simplicity, we focus in the following exposition on actor-critic methods and use Proximal Policy Optimization (PPO) (Schulman et al., 2017) for the experimental evaluation.

The underlying insight behind ITER is that at any point during RL training the latent representation of our current agent network might be significantly damaged by non-stationarity, but its outputs on the training data are comparatively unaffected (see fig. 1). Consequently, ITER aims to periodically replace the current agent network, the 'teacher', by a 'student' network which was freshly initialised and trained to mimic the teacher on the current training data. Because this re-learning and replacement step can be performed on stationary data, it allows us to re-learn a policy that matches performance on the training data but generalises better to novel test environments.

ITER begins with an initial policy $\pi^{(0)}$ and value function $V^{(0)}$ and then repeats the following steps, starting with iteration $k = 0$.

1. Use the base RL algorithm to train $\pi^{(k)}$ and $V^{(k)}$.
2. Initialise new *student* networks for $\pi^{(k+1)}$ and $V^{(k+1)}$. We refer to the current policy $\pi^{(k)}$ and value function $V^{(k)}$ as the *teacher*.
3. Distill the teacher into the student. This phase is discussed in more detail in section 4.1.
4. The student replaces the teacher: $\pi^{(k)}$ and $V^{(k)}$ can be discarded.
5. Increment $k$ and return to step 1. Repeat as many times as needed.

This results in alternating RL training with distillation into a freshly initialised student. The RL training phases continue to introduce non-stationarity until the models converge, so we want to iterate the process, repeating steps 1-4. How often we do so depends on the environment and can be chosen as a hyper-parameter. In practise we found the results to be quite robust to this choice and recommend, as general rule, to iterate as often as possible within the limits outlined in section 4.2. There, we introduce a practical implementation of ITER which re-uses data between steps 1 and 3 in order to not require additional samples from the environment.

## 4.1 Distillation Loss

Our goal during the distillation phase (step 3) is to learn a new student policy $\pi^{(k+1)}$ and value function $V^{(k+1)}$ that imitate the current teacher $(\pi^{(k)}, V^{(k)})$. If the student and teacher share the same network architecture, the student could of course imitate the teacher exactly by copying its parameters. However, since the teacher was trained under non-stationarity, its generalisation performance is likely degraded (see section 3). Consequently, we want to instead train a freshly initialised student network to mimic the teacher's outputs for the available data, but learn a better internal representation by training on a stationary data distribution collected by the teacher $\pi^{(k)}$, i.e., $s, a \sim d_{\pi^{(k)}}(s)\pi^{(k)}(a|s)$.

The student, parameterised by $\theta_{k+1}$, is trained using a linear combination of four loss terms:

$$\mathcal{L}(\theta_{k+1}) = \alpha_\pi \mathcal{L}_\pi + \alpha_V \mathcal{L}_V + \mathcal{L}_{\text{PG}} + \lambda_{\text{TD}} \mathcal{L}_{\text{TD}} \tag{1}$$

where $\lambda_{TD}$ is a fixed hyper-parameter and we linearly anneal $\alpha_\pi$ and $\alpha_V$ from some fixed initial value to zero over the course of each distillation phase.

$\mathcal{L}_\pi$ and $\mathcal{L}_V$ are supervised losses minimising the disagreement between outputs of the student and the teacher:

$$\mathcal{L}_\pi(\theta_{k+1}) = \mathbb{E}_{s \sim d_{\pi^{(k)}}} \left[ D_{\text{KL}} \left[ \pi^{(k)}(\cdot|s) \,\|\, \pi^{(k+1)}(\cdot|s) \right] \right],$$

$$\mathcal{L}_V(\theta_{k+1}) = \mathbb{E}_{s \sim d_{\pi^{(k)}}} \left[ \left( V^{(k)}(s) - V^{(k+1)}(s) \right)^2 \right]. \tag{2}$$

The additional terms $\mathcal{L}_{\text{PG}}$ and $\mathcal{L}_{\text{TD}}$ are off-policy RL objectives for updating the actor and critic:

$$\mathcal{L}_{\text{PG}}(\theta_{k+1}) = -\mathbb{E}_{s \sim d_{\pi^{(k)}}, a \sim \pi^{(k)}, s' \sim T(s,a)} \left[ \log \pi^{(k+1)}(a|s) \perp \left( \frac{\pi^{(k+1)}(a|s)}{\pi^{(k)}(a|s)} A^{(k+1)}(s,a,s') \right) \right],$$

$$\mathcal{L}_{\text{TD}}(\theta_{k+1}) = \mathbb{E}_{s \sim d_{\pi^{(k)}}, a \sim \pi^{(k)}, s' \sim T(s,a)} \left[ \left( A^{(k+1)}(s,a,s') \right)^2 \perp \frac{\pi^{(k+1)}(a|s)}{\pi^{(k)}(a|s)} \right], \tag{3}$$

where $A^{(k+1)}(s,a,s') = r(s,a) + \gamma \perp V^{(k+1)}(s') - V^{(k+1)}(s)$ denotes the estimated advantage of choosing action $a$ and $\perp$ is a `stop-gradient` operator, its operand is treated as a constant when taking derivatives of the objective. In practice, the losses in eq. (2) remain nonzero during distillation, potentially causing a drop in performance once the student replaces the teacher. The off-policy RL losses in eq. (3) allow the student to already take performance on the RL task into account, reducing or eliminating this performance drop. We use PPO losses to optimise eq. (3) in our experiments.

## 4.2 Combining Training and Distillation

To fully eliminate non-stationarity during the distillation step we need to collect additional data from the environment using a fixed teacher policy. However, this would slow down training by increasing the total number of samples required. Here, to improve sample efficiency, we propose two practical implementations of ITER which reuse data between teacher and student:

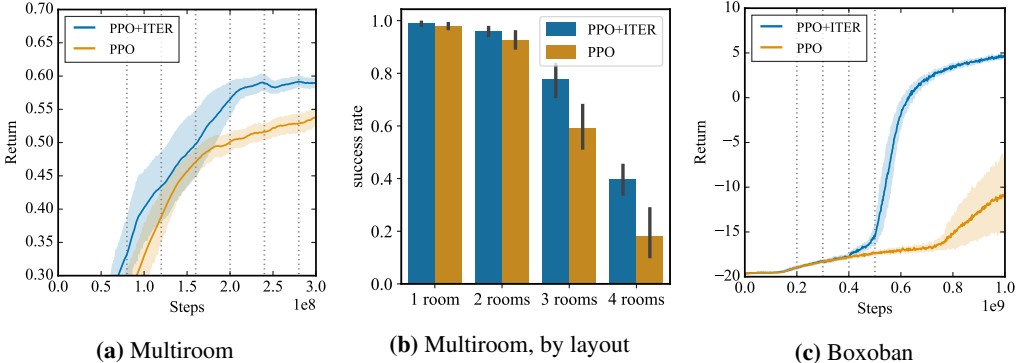

**(a)** Multiroom      **(b)** Multiroom, by layout      **(c)** Boxoban

**Figure 2:** Evaluation on *Multiroom* and *Boxoban*. Shown are mean and standard error over twelve seeds. *Left:* Return for PPO with and without ITER on *Multiroom*. Dotted lines indicate when the network was replaced by a new student. *Middle:* Evaluation on layouts with a fixed number of rooms; training is still with a random number of rooms. ITER's advantage is more pronounced for harder levels. *Right:* Return on Boxoban.

*Sequential training:* Store the last $N$ transitions that were used to update the teacher in a dataset $\mathcal{D}$. During the distillation phase, draw batches from $\mathcal{D}$ instead of collecting new data from the environment. While this does not introduce non-stationarity, it leads to evaluating the teacher on old, off-policy data, for which the quality of its outputs may be degraded. Furthermore, some of the data might be obsolete under the current state-distribution and we require additional memory to store $\mathcal{D}$.

*Parallel training:* Whenever the teacher is updated on a batch of data $\mathcal{B}$, also update the the student on the same batch. This approach introduces some non-stationarity as distillation is performed over multiple batches $\mathcal{B}$ while the teacher is changing. However, the teacher evolves much less over the course of the distillation phase than does a policy trained from scratch to achieve the same performance. In practise we found this to be a worthy trade-off. Advantages of this method are that no additional memory $\mathcal{D}$ is required, the teacher is only evaluated on data on which it is currently trained and updates to the student and teacher can be performed in parallel.

Both approaches perform similarly in our experimental validation. We use parallel training for the main experiments due to the smaller memory requirements, the ability to efficiently perform the student distillation in parallel and because tuning the hyper-parameter was significantly easier: While tuning the size of $\mathcal{D}$ for sequential ITER involves trading off overfitting (for small $\mathcal{D}$) against off-policy data (for large $\mathcal{D}$), in parallel ITER the results were robust to the choice of how many batches $\mathcal{B}$ were used in the distillation phase as long as some minimum number was surpassed. Consequently, hyper-parameter tuning for parallel ITER only involved increasing the length of the distillation phase until no drop in student performance was observed when replacing the teacher. We set $\alpha_\pi = 1$ and $\alpha_V = 0.5$ as initial values without further tuning as preliminary experiments showed no impact within reasonable ranges.

## 5 EXPERIMENTS

In the following, we evaluate ITER on *Multiroom* (Chevalier-Boisvert & Willems, 2018) and on several environments from the *ProcGen* (Cobbe et al., 2019a) benchmark which was specifically designed to measure generalisation by introducing separate test- and training levels. We also provide ablation studies showing that parallel and sequential training perform comparably, and that the loss terms eq. (3) in eq. (1) are beneficial. We find that ITER improves generalisation, which also supports our hypothesis about the negative impact of transient non-stationarity in RL. Lastly, we re-visit the SL setting from section 3 and perform additional experiments leading us to the formulation of the *legacy feature* hypothesis to explain the observed effects.

### 5.1 EXPERIMENTAL RESULTS ON MULTIROOM

First, we evaluate ITER on the *Multiroom* environment. The agent's task is to traverse a sequence of rooms to reach the goal (see fig. 6 for example layout) as quickly as possible. It can take discrete actions to rotate 90° in either direction, move forward, and open or close the doors between rooms.

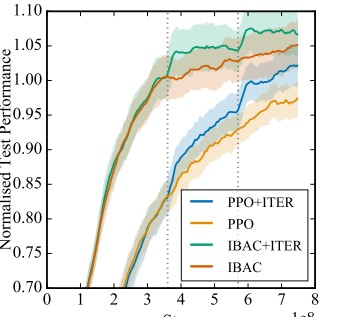 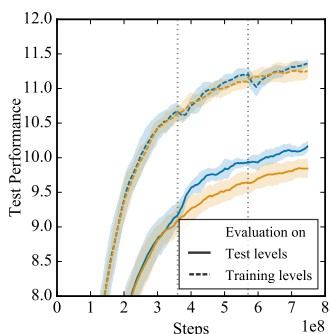 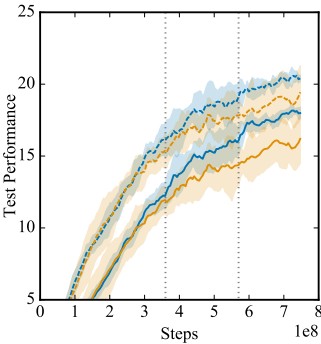

**Figure 3:** Evaluation on *ProcGen*. Dashed lines indicate replacing the teacher. *Left:* Test performance averaged over six environments (*StarPilot*, *Dodgeball*, *Climber*, *Ninja*, *Fruitbot* and *BigFish*). Shown are mean and standard error over all 30 runs (five per environment). Results are normalised by the final test-performance of the PPO baseline on each respective environment to make them comparable. We also compare against the previous state of the art method IBAC-SNI (Igl et al., 2019). *Middle:* Evaluation on *Climber*. ITER improves test performance without improving training, supporting our claim that ITER improves the latent representation of the agent. *Right:* Evaluation on *BigFish*. On some environments, ITER improves both train- and test- performance.

The observation contains the full grid, one pixel per square. Object type, including empty space and walls, as well as any object status, like direction, are encoded in the three 'colour' channels. For each episode, a new layout is generated by randomly placing between one and four connected rooms on the grid. The agent is always initialised in the room furthest from the goal. This randomness favours agents that are better at generalising between layouts as memorisation is impossible due to the high number of feasible layouts. The results are shown in fig. 2: Using ITER on top of PPO increases performance. The performance difference is more pronounced for layouts with more rooms, possibly because such layouts are harder and likely only solved later in training, at which point negative effects due to prior non-stationarity in training are more pronounced.

## 5.2 EXPERIMENTAL RESULTS ON BOXOBAN

We also evaluate ITER on *Boxoban* (Guez et al., 2018; Schrader, 2018)[1]. See fig. 8 for an example. Similarly to *Multiroom* the observation contains the full grid, one pixel per square. Object types are encoded by colour. Again, a new layout is generated at the beginning of each new episode, favouring agents that can generalise well between states. Actions allow to push, pull or move in all four cardinal directions, or do nothing. The goal of the agent is to position the four boxes, which can be pushed or pulled, on the four available targets. Walls prevent movement for both the agent and the boxes. Positive rewards are provided for positioning a box on a target ($r_b = 1$) and successfully solving each level ($r_l = 10$). A small negative reward per time-step ($r_s = -0.1$) encourages fast solutions. As shown in fig. 2, ITER learns much faster. We provide additional results and examples for wrongly chosen distillation lengths in fig. 8. Note that both for *Multiroom* and *Boxoban* we train and test on the same (very large) set of possible layout configurations, so the main expected advantage of ITER is a more sample efficient training due to better generalisation. In the next section, we will evaluate the agent on previously unseen environments, directly measuring its generalisation performance.

## 5.3 EXPERIMENTAL RESULTS ON PROCGEN

Next, we evaluate ITER on several environments from the *ProcGen* benchmark. We follow the evaluation protocol proposed in (Cobbe et al., 2019a): for each environment, we train on 500 randomly generated level layouts and test on additional, previously unseen levels. Due to computational constraints, we consider a subset of five environments. We chose *StarPilot*, *Dodgeball*, *Climber*, *Ninja*, *Fruitbot* and *BigFish* based on the results presented in (Cobbe et al., 2019a) as they showed baseline generalisation performance better than a random policy, but with a large enough generalisation gap. ITER improves performance for both PPO and Information Bottleneck Actor Critic (IBAC) with selective noise injection (Igl et al., 2019). Results are presented in fig. 3 and more individual plots, including performance on training levels, can be found in the appendix. In fig. 4 we show in ablations

---

[1]Our simplified `Boxoban-Train-v0` also allows pulling boxes to reduce computational costs

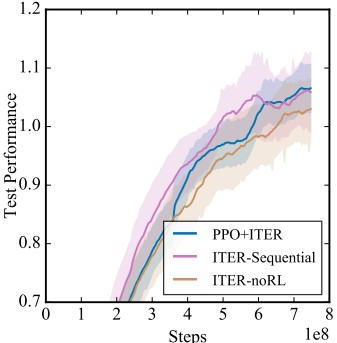 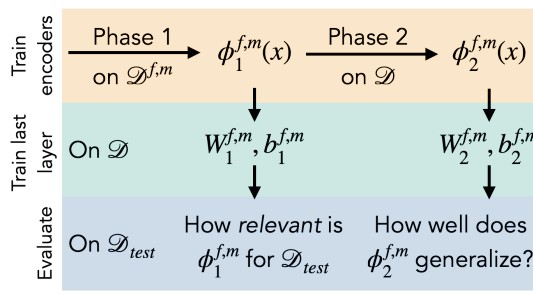

**Figure 4:** *Left:* Ablation studies with sequential ITER and ITER without terms $\mathcal{L}_{\text{PG}}$ and $\mathcal{L}_{\text{TD}}$ (eq. (3)). *Right:* Schematic depiction of training setup for fig. 5 (middle and right). More details are given in section 5.4. $\mathcal{D}$ is the unmodified CIFAR-10 training data while for $\mathcal{D}^{f,m}$ modification $m \in \{\texttt{Noisy Labels}, \texttt{Wrong Labels}, \texttt{Dataset Size}\}$ is applied to the fraction $(1-f)$ of all data-points. In this two phase training setup, we first train on $\mathcal{D}^{f,m}$ during phase 1 and continue on $\mathcal{D}$ during phase 2. A linear predictor parameterised by $(W_i^{f,m}, b_i^{f,m})$ is trained on $\mathcal{D}$ after each phase $i$, while holding the encoder $\phi_i^{f,m}(x)$ fixed. Evaluation of the resulting classifiers is performed on the original test data. Classifier $i = 1$ measures the relevance of the legacy features while classifier $i = 2$ measure the final generalisation performance.

that both the parallel and sequential implementations of ITER perform comparably, while not using the off-policy RL terms $\mathcal{L}_{\text{PG}}$ and $\mathcal{L}_{\text{TD}}$ in eq. (1) decreases performance.

Similarly to previous literature (Cobbe et al., 2019b; Igl et al., 2019), we found that weight decay improves performance and apply it to all algorithms evaluated on *ProcGen*. Our results show that the negative effects of non-stationarity cannot easily be avoided through standard network regularisation: we can improve test returns through ITER despite regularisation with weight decay and IBAC, both shown to be among the most effective regularisation methods on this benchmark (Igl et al., 2019).

In the previous two sections we have shown the effectiveness of ITER in improving generalisation of RL agents. Because the main mechanism of ITER is in reducing the non-stationarity in the training data which is used to train the agent, this result further supports that such transient non-stationarity is detrimental to generalisation in RL.

## 5.4 SUPERVISED LEARNING ON CIFAR-10

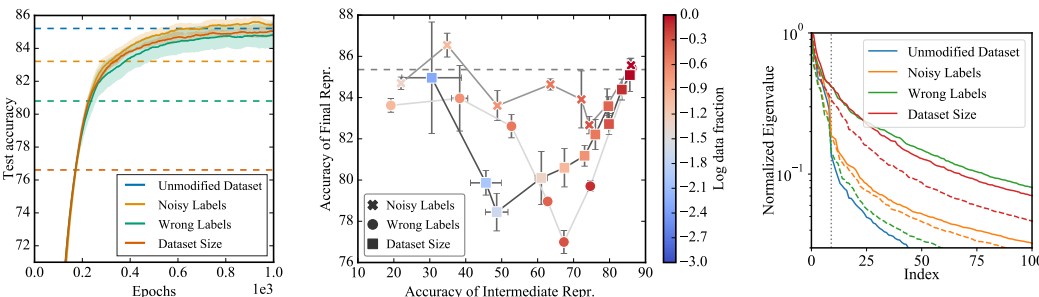

**Figure 5:** *Left:* Test accuracy of students (solid lines) that only learn to mimic the behaviour of poorly generalising teachers in fig. 1 (dashed lines). *Middle:* Final test accuracy of networks trained consecutively on two different datasets. The $x$-axis shows the accuracy of using encoders trained on the first dataset, retraining only the last layer on the second: nearly useless earlier representations impact future learning much less than slightly sub-optimal ones. Markers indicate modifications to first dataset; colours indicate the fraction of unmodified data points $f$. Dashed line shows accuracy for $f = 1$. *Right:* Singular values of feature matrix $\Phi$, normalised by the largest SV. Solid lines show intermediate values of $f$ with low test accuracy, dashed lines small values of $f$ with higher accuracy. More plots can be found in the appendix.

In this section, we aim to further understand the mechanism by which non-stationarity impacts generalisation by revisiting the easily controlled SL setting presented in section 3.

First, we confirm that, like with ITER for RL, we can improve generalisation while only learning to mimic the outputs of a poorly generalising teacher for the training data. We train the teacher using the same setup as in section 3. In a second step, we train a freshly initialised student for 1000 epochs to fit the argmax predictions of this teacher on the training data, i.e., the true labels in the training data are unused. Test accuracy is still measured using the true labels. The results of this distillation phase are shown in fig. 5 (left). The student (solid lines) recovers the test accuracy achieved by stationary training, compared to the poor asymptotic teacher performance (dashed lines) from fig. 1. This confirms that the teacher's outputs on the training data are suitable targets for distillation.

Second, we aim to better understand why non-stationarity affects the generalisation performance. To do so, we investigate the latent representation of the network. We view all but the last layer as the encoder, producing the latent *representation* $\phi(x) \in \mathbb{R}^p$ for input $x$, on which the classification is linear: $y = \text{softmax}(W\phi(x) + b)$ with $W \in \mathbb{R}^{|C| \times p}$ denoting a weight matrix and $b \in \mathbb{R}^{|C|}$ denoting a bias vector. By *features* we refer not to the representation $\phi(x)$ as a whole, but to aspects of the input to which the encoder learned to pay attention and which therefore can impact the latent representation (Kriegeskorte & Kievit, 2013). More quantitatively, we can define the representation matrix $\Phi \in \mathbb{R}^{N \times p}$, consisting of the latent representations of all $N = 10000$ test data points. Performing Singular Value Decomposition (SVD) on $\Phi$ yields mutually orthogonal directions (the right-singular vectors) in which the latent representations of the various inputs are different from one another. We can see each such direction as corresponding to one feature, with the corresponding singular value expressing its strength, i.e., how strongly it impacts the latent representation.

Our hypothesis is that under a non-stationary data distribution, the encoder is more likely to reuse previously learned features (as these are already available) instead of learning new features from scratch. If these old (or 'legacy') features generalise worse on the new data, for example because they are overfit to a smaller or less diverse dataset, this in turn deteriorates generalisation permanently if they are not replaced. This leads to two predictions: First, the observed drop in generalisation should only occur if the previously learned features are relevant for the new task, but suboptimal. If they are irrelevant, they will not be reused. If they are optimal, they do not negatively impact generalisation. Second, we expect the final network to rely on more features in its latent representation if it is reusing suboptimal features: because these are not as general, more features are required to discriminate between all inputs.

To experimentally evaluate both predictions, we simplify the experimental setting to two phases (see fig. 4 for a schematic depiction of the training setup). The first training phase uses a stationary, but modified, dataset $\mathcal{D}^{f,m}$, and the second phase uses the full, unmodified, training dataset $\mathcal{D}$. To generate $\mathcal{D}^{f,m}$, we use the same modifications as before, $m \in \{\texttt{Noisy Labels}, \texttt{Wrong Labels}, \texttt{Dataset Size}\}$, but instead of annealing the fraction of correct data points $f$ from 0 to 1 as in section 3, it is fixed at a certain value. Changing this value $f$ allows us to tune the relevance of the features learned on $\mathcal{D}^{f,m}$ (also see fig. 6). In this setup the only non-stationarity is the change in data from phase 1 to phase 2. We first train the network for 700 epochs on $\mathcal{D}^{f,m}$, which yields an intermediate representation $\phi^{f,m}_{inter}(x)$, followed by another 800 epochs on $\mathcal{D}$ yielding the final representation $\phi^{f,m}_{final}(x)$ (see fig. 7 for training curves).

To test our first hypothesis, we want to measure how relevant the representation $\phi^{f,m}_{inter}(x)$ is for the final data distribution and how well $\phi^{f,m}_{final}(x)$ generalises. We train a linear predictor for each *fixed* representation on the *full* dataset $\mathcal{D}$.[2] The test accuracy of the classifier based on $\phi^{f,m}_{inter}(x)$ measures how well we can perform on $\mathcal{D}$ with features learned on $\mathcal{D}^{f,m}$, i.e., their *relevance*. The test accuracy of the classifier based on $\phi^{f,m}_{final}$ measures how well the final network was able to recover from the initial bias and learn to generalise well despite non-stationarity.

In fig. 5 (middle), we plot the accuracy of the intermediate predictor (i.e., the relevance) on the $x$-axis and of the final generalisation performance on the $y$-axis. Each point corresponds to one value of $f \in (0, 1]$, shown as colour, and one modification type $m$ as indicated by the marker shape. By changing $f$ from 0 to 1 (i.e., from blue to dark red) we can increase the relevance of the intermediate features from nearly irrelevant to optimal on $\mathcal{D}$. Interestingly, an almost useless intermediate representation (30% on the $x$-axis) does not impede the final performance much, while relevant but suboptimal intermediate features (around 60% on the $x$-axis) lead to a marked drop in

---

[2] The linear predictor is $y = \sigma(W\phi_{f,m}(x) + b)$, where $\sigma$ is the softmax function and $x$ the input image.

performance. This supports our first hypothesis. The strong final performance for $f \rightarrow 0$ (i.e., for low relevance) also rules out decreased network flexibility, for example due to dead neurons (for ReLUs) or saturation (for tanh), as the main driver of reduced generalisation.

Our second prediction is that relevant, but suboptimal features in $\phi_{inter}^{f,m}$ should lead to the usage of more features in $\phi_{final}^{f,m}$ compared to irrelevant or optimal features. To test this prediction, we plot in Figure 5 (right) the singular values (SVs) of the representation matrix $\Phi \in \mathbb{R}^{N \times p}$ as defined above. We plot the values for the smallest values of $f$ (dashed lines, "irrelevant features") and for intermediate values of $f$ (solid lines, 'relevant but sub-optimal features'). The blue line corresponds to optimal features. The tails of the singular values are heavier for intermediate values of $f$, indicating that the network is relying on more features in those cases, supporting our second prediction.

## 6 RELATED WORK

Knowledge distillation (Hinton et al., 2015) with identical teacher and student architectures has been shown to improve test accuracy (Furlanello et al., 2018), even in the absence of non-stationarities in the data. This improvement has been attributed to the ease of predicting the output distribution of the teacher compared to the original 'hard' labels (Mobahi et al., 2020; Gotmare et al., 2019). While we apply such 'soft' distillation for ITER on RL, we use 'hard' labels in our SL experiments.

Policy distillation has been applied to RL (Czarnecki et al., 2019), for example for multi-task learning and compression (Teh et al., 2017; Rusu et al., 2015; Parisotto et al., 2016), imitation learning (Ross et al., 2011), or faster training (Schmitt et al., 2018; Ghosh et al., 2018). Closer to ITER, Czarnecki et al. (2018) use policy distillation to learn a sequence of agents. However, their Mix & Match algorithm solves tasks of growing complexity, for example, to grow the action space of the agent, not to tackle generalisation or non-stationarity.

While the topic of non-stationarity is central to the area of continual learning (see (Parisi et al., 2019) for a recent review), the field is primarily concerned with preventing catastrophic forgetting (French, 1999) when the environment or task changes during training (Li & Hoiem, 2017; Schwarz et al., 2018). For non-stationary environments during agent deployment, the approach is typically to detect such changes and respond accordingly (Choi et al., 2000; Da Silva et al., 2006; Doya et al., 2002). By contrast, we assume a stationary environment and investigate the impact of transient non-stationarity, for example induced by an improving policy. We also show that intentionally forgetting the representation, but not the learned outputs, can improve generalisation in this case.

Neural networks are used in deep RL to allow generalisation across similar states (Sutton & Barto, 2018). Several possibilities have been proposed to further improve generalisation, including to provide more diverse training environments (Tobin et al., 2017), inject noise into the environment (Stulp et al., 2011; Lee et al., 2020), incorporate inductive biases in the architecture (Kansky et al., 2017), or regularise the network (Cobbe et al., 2019b; Igl et al., 2019; Liu et al., 2019). While regularisation reduces overfitting, we show in our experiments that this is insufficient to counter the negative effects of non-stationarity, and that ITER can be complementary to other types of regularisation.

## 7 CONCLUSION

In this work, we investigate the impact of non-stationarity on the generalisation performance of trained RL agents. First, in several SL experiments on the CIFAR-10 dataset, we confirm that non-stationarity can considerably degrade test performance while leaving training performance nearly unchanged. To explain this effect, we propose and experimentally support the *legacy feature* hypothesis that networks exhibit a tendency to adopt, rather then forget, features learned earlier during training if they are sufficiently relevant, though not necessarily optimal, for the new data. We also show that self-distillation, even without using the true training labels, improves performance on the test-data.

Many deep RL algorithms induce similar transient non-stationarity, for example due to a gradually converging policy. Consequently, to improve generalisation in deep RL, we propose ITER which reduces the non-stationarity the agent networks experience during training. Our experimental results on the *Multiroom* and *ProcGen* benchmarks empirically support the benefits of ITER, indicating that transient non-stationarity has a negative impact in deep RL.

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

## A  PSEUDO CODE

---

**Algorithm 1:** Pseudo-Code for parallel ITER

---

1 **Input** Length of initial RL training phase $t_{init}$, length of distillation phase $t_{distill}$

2 **Initialise** $k \leftarrow 0$, policy $\pi^{(k)}$, value function $V^{(k)}$

3 *// Normal RL training at the beginning*
4 **for** $t_{init}$ steps **do**
5 $\quad$ $\mathcal{B} \leftarrow$ collect trajectory data using $\pi^{(0)}$
6 $\quad$ Update $\pi^{(0)}$ and $V^{(0)}$ using standard RL method using $\mathcal{B}$

7 *// Combine further RL training of $\pi^{(k)}, V^{(k)}$ with distillation of $\pi^{(k+1)}, V^{(k+1)}$*
8 **while** not converged **do**
9 $\quad$ **Initialise** student policy $\pi^{(k+1)}$ and value function $V^{(k+1)}$
10 $\quad$ **for** $t_{distill}$ steps **do**
11 $\quad\quad$ $\alpha_V, \alpha_\pi \leftarrow$ linear annealing to 0 over $t_{distill}$ steps
12 $\quad\quad$ $\mathcal{B} \leftarrow$ collect trajectory data using $\pi^{(k)}$
13 $\quad\quad$ Update $\pi^{(k)}$ and $V^{(k)}$ with standard RL method using $\mathcal{B}$
14 $\quad\quad$ Update $\pi^{(k+1)}$ and $V^{(k+1)}$ with eq. (1) using $\mathcal{B}, \alpha_V, \alpha_\pi, \pi^{(k)}$ and $V^{(k)}$

15 $\quad$ *// Housekeeping*
16 $\quad$ Discard $\pi^{(k)}$ and $V^{(k)}$
17 $\quad$ Set $k \leftarrow k + 1$

---

## B  SUPERVISED LEARNING

**Table 1:** Numerical values of results presented in fig. 1. The 'Rel' column shows the error normalised by the error of the unmodified dataset. The error on the test-data deteriorates worse than on the training data, not only in absolute, but also relativ terms.

**Table 2:** Hyper-parameters used in the supervised learning experiment on CIFAR-10

| | Training Error in % | Rel. | Testing Error in % | Rel. |
|---|---|---|---|---|
| Unmodified | $0.17 \pm 0.09$ | 1.0 | $14.8 \pm 0.70$ | 1.0 |
| Noisy Labels | $0.19 \pm 0.09$ | 1.13 | $16.8 \pm 0.70$ | 1.14 |
| Wrong Labels | $0.20 \pm 0.08$ | 1.22 | $19.2 \pm 0.43$ | 1.30 |
| Dataset Size | $0.18 \pm 0.08$ | 1.05 | $23.4 \pm 0.83$ | 1.58 |

| Hyper-parameter | Value |
|---|---|
| SGD: Learning rate | $3 \times 10^{-4}$ |
| SGD: Momentum | 0.9 |
| SGD: Weight decay | $5 \times 10^{-4}$ |

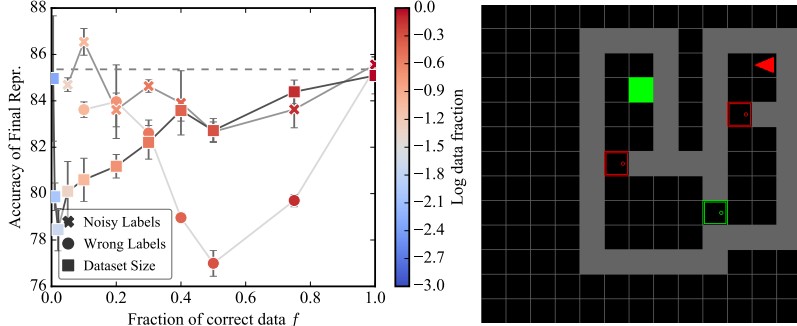

**Figure 6:** *Left:* Same results as in fig. 5 (middle), but with the fraction of correct data points $f$ on the x-Axis. *Right:* Multiroom example layout. The red agent needs to reach the green square, avoiding walls (grey) and passing through doors (blocks with coloured outline).

Here we provide additional training details and results for the supervised learning experiments performed on the CIFAR-10 dataset. We used a ResNet18 architecture without Batchnorm, hyper-parameters for the SGD optimiser are given in table 2. In table 1 we provide exact numerical values for the results in fig. 1. We also provide values for the *relative* change in error rate due to the introduction of non-stationarities, for which the test-performance is also more affected than the train performance.

In fig. 6, we show the same results as in fig. 5, but here showing the $f$ values used to generate $\mathcal{D}_{f,m}$ on the x-Axis. The same 'dips' in performance are visible, however from this figure it is clear that Dataset Size experiences it for much smaller values of $f$, which is unsurprising, giving the missing influences of a diverse input-data distribution.

Lastly, in fig. 7, we provide the individual training runs used to generate fig. 5(middle) and fig. 6.

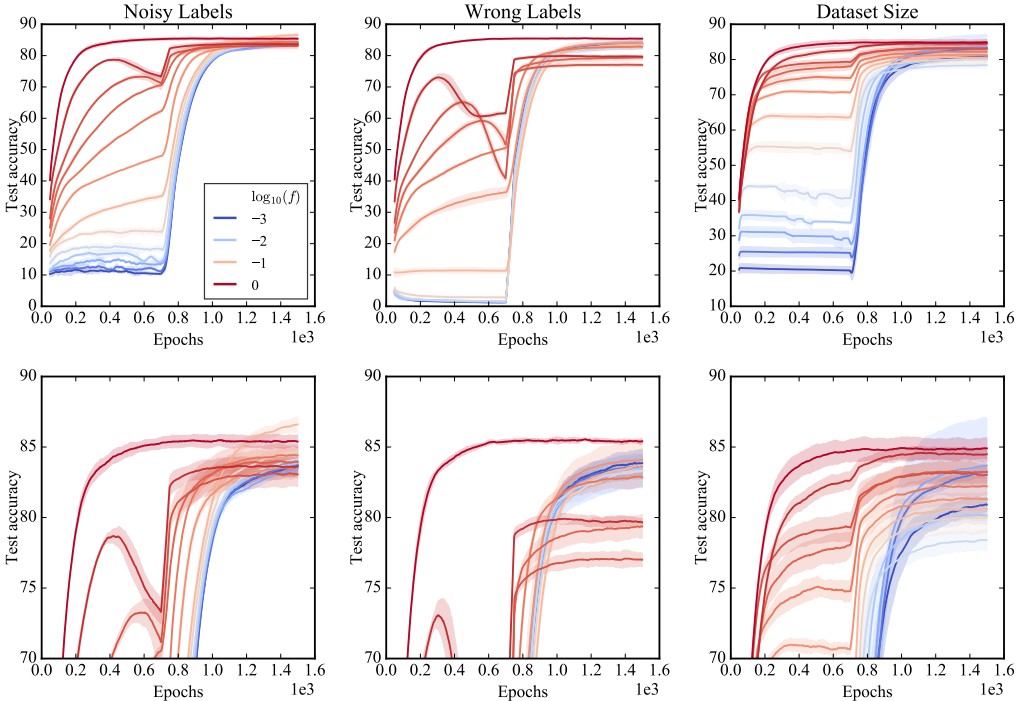

**Figure 7:** Individual training curves for the data used in fig. 5(middle) and fig. 6. The bottom row shows the same data as the top row, just 'zoomed in'.

### B.1 MULTIROOM

In table 3 we show the hyper-parameters used for the Multiroom experiments which are shared between 'PPO' and 'PPO+ITER'. We note that our Multiroom environment uses the same modification that was used in (Igl et al., 2019) to make the environment fully observable. In the original environment, the agent only observed its immediate surrounding from an ego-centric perspective, thereby naturally generalising across various layouts. Instead, full observability introduces the need to *learn* how to generalise. Our network consists of a three layer CNN With 16, 32 and 32 filters respectively, followed by a fully connected layer of size 64. One max-pooling layer is used after the first CNN layer. We use $t_{init} = 4 \times 10^7$ and $t_{distill} = 4 \times 10^7$ (see algorithm 1) for the duration of the initial RL training phase and the following distillation phases.

### B.2 BOXOBAN

For Boxoban, we re-use the same architecture and hyper-parameters as for Multiroom, but with a reduced learning rate ($1.0e - 04$) in order to stabilise training.

**Table 3:** Hyper-parameters used for Multiroom

| Hyper-parameter | Value |
| --- | --- |
| PPO: $\lambda_{\text{Entropy Loss}}$ | 0.01 |
| PPO: $\lambda_{\text{TD}}$ | 0.5 |
| PPO: $\epsilon_{\text{Clip}}$ | 0.2 |
| PPO Epochs | 4 |
| PPO Minibatch Size | 2048 |
| Parallel Environments | 32 |
| Frames per Env per Update | 256 |
| $\lambda_{GAE}$ | 0.95 |
| $\gamma$ | 0.99 |
| Adam: Learning rate | $7 \times 10^{-4}$ |
| Adam: $\epsilon$ | $1 \times 10^{-5}$ |

**Table 4:** Hyper-parameters used for ProcGen

| Hyperparameter | Value |
| --- | --- |
| PPO: $\lambda_{\text{Entropy Loss}}$ | 0.01 |
| PPO: $\lambda_{\text{TD}}$ | 0.5 |
| PPO: $\epsilon_{\text{Clip}}$ | 0.2 |
| PPO Epochs | 3 |
| PPO Nr. Minibatches | 8 |
| Parallel Environments | 64 |
| Frames per Env per Update | 256 |
| $\lambda_{GAE}$ | 0.95 |
| $\gamma$ | 0.999 |
| Adam: Learning rate | $5 \times 10^{-4}$ |
| Adam: $\epsilon$ | $1 \times 10^{-5}$ |
| Adam: Weight decay | $1 \times 10^{-4}$ |

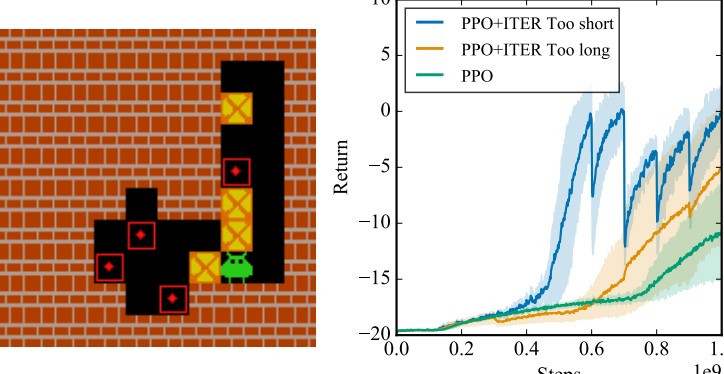

**Figure 8:** *Left:* Boxoban example layout. The green agent needs to push (or pull) yellow boxes on the red targets, avoiding walls. *Right:* Additional Boxoban results showing consequences of choosing a wrong distillation length, either too short or too long. Note that `ITER Too short` uses the same distillating length as the results in fig. 2, but continues with distillation past $0.5e9$ steps.

In fig. 8 we show results for wrongly chosen distillation lengths. `Too short` uses the same distillation length as the main results in fig. 2, but continues distilling until the end of training. Because after $0.5e9$ steps the policy performance and complexity increase considerably, a longer distillation period would be required. On the other hand, using a distillation period twice as long from the start (`Too long`) leads to slower training. We did not experiment with increasing the distillation length over the course of training, since in our experiments earlier stopping of ITER was sufficient for optimal performance.

## B.3 PROCGEN

In fig. 9 we show all results on the various *ProcGen* environment from which the summary plots in the main text (figs. 3 and 4) are computed. We use the same (small) IMPALA architecture as used by (Cobbe et al., 2019b). Training is done on 4 GPUs in parallel. One GPU is continuously evaluating the test performance, the other three are used for training. Their gradients are averaged at each update step. The hyper-parameters given in table 4 are *per GPU*. The $x$-Axis in fig. 9 shows the total number of consumed frames, i.e. $250 \times 10^6$ per *training* GPU. The distillation phase takes $t_{distill} = 70 \times 10^6$ frames (again per GPU) and we linearly anneal $\alpha_\pi$ from 1 to 0 and $\alpha_V$ from 0.5 to 1. The values of $\alpha_\pi$ and $\alpha_V$ were chosen to reflect the relative weight between $\mathcal{L}_{PG}$ and $\mathcal{L}_{TD}$ in eq. (1) and no further tuning was done. The initial RL training phase takes $t_{init} = 50 \times 10^6$ frames. The distillation length was chosen based on preliminary experiments on *BigFish* by increasing its

length in steps of $10 \times 10^6$ frames until no drop in training performance was experienced when switching to a new student.

Due to the high computation costs of running experiments on the *ProcGen* environment (4 GPUs for about 24h for each run), we decided to exclude environments from the original benchmark based on results presented by Cobbe et al. (2019a), figures 2 and 4. We excluded environments for two different reasons, either because the generalisation gap was small (Chaser, Miner, Leaper, Boss Fight, Fruitbot) or because generalisation did not improve at all during training after a very short initial jump (CaveFlyer, Maze, Heist, Plunder, Coinrun), indicating that either it was too hard, or a very simple policy already achieved reasonable performance.

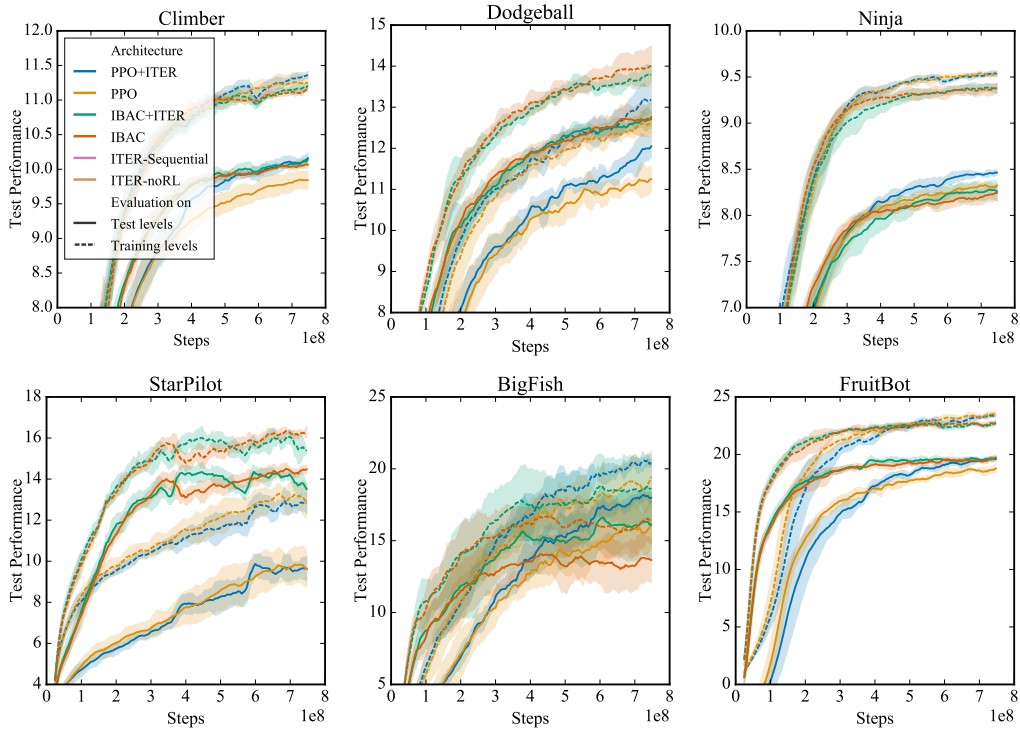

**Figure 9:** All individual results on *ProcGen*. Shown is the mean and standard deviation across two random seeds.

