# OpenReview forum: "Transient Non-stationarity and Generalisation in Deep Reinforcement Learning"
_ICLR.cc/2021/Conference — ICLR 2021 Poster_

### Official Review · AnonReviewer2 · 2020-10-18
**interesting paper analyzing how non-stationarity affects generaliztion**

**Rating:** 8
**Confidence:** 4

**Review:**

This paper presents empirical evidence that non-stationarity data typical in deepRL settings can affect the intermediate representation of deep neural network and affect testing performance. The paper is easy to read and the authors provide experiments to support the their observations and claims. Overall I think this is a good paper and in the following I suggest some good to have additions.

(1) The examples for the supervised learning setting clearly demonstrates the impact of non-stationary data. However, given that this is inspired by the problems under DRL setting, it will be interesting to do more analysis of this effect on some DRL tasks. For example, an analysis for offline RL might be a good setting to study this effect.

(2)Imitation learning algorithm like Dagger might be another good example to demonstrate the effect of nonstationarity. The data under the Dagger setting is also changing overtime and it will be interesting to see how it affects the student policy.

(3) The RL experiment is mainly done in the on policy (PPO) settings. Some experiments with off policy RL setting might be useful, and the effect of the non-stationarity might be more pronounced as well.

---

> ### Author Response · Authors · 2020-11-19
> **Thank you for your review**
>
> Thank you for your review.
>
> We agree that off-policy and especially offline RL is both an interesting application and might facilitate more in-depth analysis by allowing one to modify the state-distribution, similar to our SL experiments and we are planning to explore this in future work.
>
> Thank you also for the imitation learning suggestion! It’s something we hadn’t considered so far.

---

### Official Review · AnonReviewer1 · 2020-10-20
**This paper tackles the problem of distribution shift in the context of reinforcement learning, e.g. resulting from gradual exploration of the state-space or optimization of the critic. The main idea is that this kind of non-stationarity can have a lasting negative effect on generalization, even after the data distribution has converged. The paper proposes a solution which is based on iteratively distilling the current model into a new model that is optimized from scratch.**

**Rating:** 7
**Confidence:** 4

**Review:**

The paper deals with a relevant issue. The simplified supervised learning setting is a good way of looking at the issue of non-stationarity in isolation and it makes a compelling case that neural networks optimized by SGD can have generalization issues in settings where the data distribution changes over time, even after the data distribution converges. The solution proposed by the paper is simple and can be applied to most off-the-shelf RL algorithms. My main criticism would be that the hybrid objective feels rather ad-hoc and that the proposed method could use a bit more theoretical justification. Also, since the issue being tackled here is quite general to RL, a wider set of benchmarks and base-algorithms (in addition to PPO) would be necessary to get a better picture.
Regardless, I believe the community would benefit from the inclusion of this paper.

### Pros

* The paper is well-written. The story is easy to follow and well-motivated.
* The central problem that is being tackled is highly relevant.
* I like that the problem is illustrated in a supervised-learning setting, which allows to investigate without the noise of a typical RL setup.
* The proposed modification is simple and can be applied to a wide-range of algorithms.
* The "legacy feature" effect is an interesting phenomenon and the additional experiments inspecting it are useful.

### Cons

* In Figure 5, the middle panel is a bit hard to parse. I don't know how to improve this but in its current state, too many variables are being presented at once.
* While the method being presented is quite simple, it also seems a bit ad-hoc. For instance, it would be nice if the teacher-student distillation could be supported by some convergence guarantees, e.g. by showing that (under some circumstances) replacing the teacher by the student does not increase the loss.
* The set of benchmarks considered in the paper is a good proof-of-concept, but additional experiments with different environments and base-algorithms is necessary to better judge the merits of the approach.

---

> ### Author Response · Authors · 2020-11-19
> **Thank you for your review**
>
> Thank you for your review!
>
> We agree that Figure 5 (middle) is quite packed with information. We updated the corresponding description in the text and added a schematic figure for the experimental procedure, hoping it provides more clarity. We’re very grateful for any additional feedback/ideas.
>
> > While the method being presented is quite simple, it also seems a bit ad-hoc. For instance, it would be nice if the teacher-student distillation could be supported by some convergence guarantees, e.g. by showing that (under some circumstances) replacing the teacher by the student does not increase the loss.
>
> Thank you for the suggestion, it’s something we plan to explore in future work. We expect it to be quite tricky as the observed effect of non-stationarity wouldn’t be present for simple (e.g. linear) function approximators while generalization in typical neural networks isn’t well understood yet.
>
> > The set of benchmarks considered in the paper is a good proof-of-concept, but additional experiments with different environments and base-algorithms is necessary to better judge the merits of the approach.
>
> We have added an additional experiment on the Sokoban environment, which also utilizes procedural generation of levels.

---

> > ### Comment · AnonReviewer1 · 2020-11-24
> > **Reply to the authors**
> >
> > Thank you very much for your reply. I maintain my original score and my position that the paper should be accepted.
> >
> > To make it clear what would make me increase my score:
> >
> > * Given the breadth of settings that PPO has been applied to in the literature, the experiments could be more extensive in my opinion, though I appreciate the additional results on sokoban. What I find missing here are some continuous settings like the ones found in gym-mujoco, the deepmind control suite, roboschool or D4RL.
> >
> > * A more grounded justification of the distillation objective.
> >
> > I believe either of these would make the paper stronger.

---

### Official Review · AnonReviewer3 · 2020-10-28
**Official Blind Review #3**

**Rating:** 5
**Confidence:** 3

**Review:**

This paper investigates an interesting problem that transient non-stationarity can affect the generalization of the neural network. This paper first conducts experiments on a supervised learning task to illustrate that transient non-stationarity can lead to degenerated performance on testing set. Then, the paper proposes an RL algorithm called ITER to avoid the negative impact of such non-stationarity.

Strengths:
+ This paper observes a novel problem that may appear in RL and designs a new algorithm to prevent such a problem.
+ This paper investigates this problem through experiments on supervised learning tasks.

Weaknesses:
- The new algorithm ITER doubles the computational costs.
- Experiments on Page 7 are trying to further illustrate the mechanism for such a phenomenon. However, this subsection is not very clear.
- Experiment results on Procgen is fair but not significant.

Minor comments:
- In Section 2, the advantage function is defined as A^\pi(s,a,s') but later used as A^\pi(s,a) without additional explaination.
- f in $\mathcal{D}_{f,m}$ is not explained when it first appears. I guess it is the ratio of the modification.

---

> ### Author Response · Authors · 2020-11-19
> **Thank you for your review**
>
> Thank you for your review! We provide some clarification below.
>
> > The new algorithm ITER doubles the computational costs.`
>
> While this is indeed the case (however only during the distillation phase!), this additional computation can be performed in parallel and as such the runtime is not doubled. We also do not require any additional environmental steps, which is often the more important bottleneck: in many scenarios, some extra computation by the learner is likely a small price to pay for improved generalisation. In our experiments, the runtime was about 1.3-1.5 times as high for the large ProcGen models, which we expect could be brought down even further by running the distillation step on a separate GPU.
>
> > Experiment results on Procgen is fair but not significant.
>
> The magnitude of improvements is in a comparable range to other methods that are evaluated on the ProcGen benchmarks (e.g. Igl et al.: Generalization in RL with Selective Noise Injection; Lasking et al.: RL with Augmented Data; Raileanu et al.: Automatic Data Augmentation for Generalization in DLR; Cobbe et al.: Phasic Policy Gradient; Jiang et al.: Prioritized level replay).
> The absolute magnitude of improvement that is possible is therefore as much a feature of the environment as of the algorithm: generalization and its improvement on ProcGen are hard. The important point is that ITER does improve generalization, therefore providing evidence that the discussed non-stationarity impacts generalization.
> Also please note that the magnitude of improvement is greater on Multiroom and Sokoban [results added in paper revision] , which are complex environments (despite being visually slightly simpler).
>
> > Experiments on Page 7 are trying to further illustrate the mechanism for such a phenomenon. However, this subsection is not very clear.
>
> We have updated this section (including a new schematic figure explaining the training setup)  and are grateful for any additional feedback on clarity or how it could be improved.
>
> Thank you for your other comments, we’ve incorporated them in the updated draft.

---

### Official Review · AnonReviewer4 · 2020-10-29
**Limited understanding on the non-stationarity in RL and benefit from ITER**

**Rating:** 5
**Confidence:** 5

**Review:**

The paper proposes a mechanism of re-learning (ITER; iterative re-learning) to handle issue from non-stationarity in RL. Some gain of using the proposed method is experimentally presented. My major concern is the limited understanding of the non-stationarity issue and ITER.

Pros)
- The authors propose a new approach to resolve the non-stationarity issue in RL. The intuition itself makes a sense as human often performs such relearning processes to escape from the local minimum.

- Some gain of ITER is empirically demonstrated in various setting.

Cons)
- Although the underlying intuition sounds delightful, the understanding on the non-stationarity is insufficient. In particular, the authors design a setup of "supervised" learning to show importance of non-stationarity. However, it is a weak evidence on the importance of non-stationarity in "reinforcement" learning. Indeed, the gain of using ITER is not significant.

- The authors do not clearly define the generalization which the relearning aims to improve. It may depend on the neural network architecture and environment and so on. However, presuming that the paper reports the best gain possible, it is hard to accept that ITER always improves generalization.

- I have a concern on the little gain, which is shown by the comparison between PPO vs. PPO+ITER with the same hyperparameters. A fair comparison may use the best hyerparameters for each of PPO and PPO+ITER. Or, at least, there need comparisons with different hyperparameters to claim consistent improvement.

---

> ### Author Response · Authors · 2020-11-19
> **Thank you for your review**
>
> Thank you for your review! We provide some clarifications below.
>
> > The authors do not clearly define the generalization which the relearning aims to improve. It may depend on the neural network architecture and environment and so on. However, presuming that the paper reports the best gain possible, it is hard to accept that ITER always improves generalization.
>
> We refer to generalisation to environment states not yet seen during training (we emphasise this now more in the beginning of section 4). Unlike many commonly used RL benchmarks, our experiments rely on procedural generation of tasks. This creates a sufficient diversity of states to make generalisation a meaningful performance bottleneck, and in the case of ProcGen allows us to explicitly measure a generalisation gap by separating levels used for training and testing, similar to the use of a validation set in supervised learning.
>
> We expect our approach to improve performance in environments where such generalisation is important, either because testing encounters states that are guaranteed to be unseen (like ProcGen) or the agent is faced with a very large number of diverse states (like Multiroom, or Sokoban [results added in paper revision] in which each episode has a different layout).
>
> Good generalisation can also become important when diverse states arise from noise in transition dynamics, or from broad starting state distributions. However, we wouldn’t expect our method to improve performance in near deterministic environments like Atari games.
>
> > Although the underlying intuition sounds delightful, the understanding on the non-stationarity is insufficient. In particular, the authors design a setup of "supervised" learning to show importance of non-stationarity. However, it is a weak evidence on the importance of non-stationarity in "reinforcement" learning. Indeed, the gain of using ITER is not significant.
>
> Our fundamental insight is that certain kinds of non-stationarity interfere with the SGD training of neural networks and can degrade their generalization performance, which should be independent of whether the network is used in a SL or RL setup.
> This is easiest shown in a supervised learning setup as it has fewer confounding factors. Furthermore, there is no reason this should not, in principle, translate to RL. The improved performance through ITER provides evidence that this is indeed the case and non-stationarity in RL is impeding generalization (at least in environments such as those described above).
>
> We agree that research aiming at better understanding of which type of non-stationarity in RL is the main problem for generalization is an exciting direction for future work. However, we believe this is out of scope of this paper as such research is quite challenging and we already show (a) that such a problem exists and (b) how to alleviate it during training, which we believe is a substantial contribution.
> Further ablation studies in RL are tricky because independently modifying for example the state distribution, causes the policy updates to be off-policy, thereby introducing significant confounding factors.
>
> The magnitude of improvements is in a comparable range to other methods that are evaluated on the ProcGen benchmarks (e.g. Igl et al.: Generalization in RL with Selective Noise Injection; Lasking et al.: RL with Augmented Data; Raileanu et al.: Automatic Data Augmentation for Generalization in DLR; Cobbe et al.: Phasic Policy Gradient; Jiang et al.: Prioritized level replay).
> The absolute magnitude of improvement that is possible is therefore as much a feature of the environment as of the algorithm: generalization and its improvement on ProcGen are hard. The important point is that ITER does improve generalization, therefore providing evidence that the discussed non-stationarity impacts generalization.
> Also please note that the magnitude of improvement is greater on Multiroom and Sokoban, which are complex environments (despite being visually slightly simpler).
>
> > I have a concern on the little gain, which is shown by the comparison between PPO vs. PPO+ITER with the same hyperparameters. A fair comparison may use the best hyerparameters for each of PPO and PPO+ITER. Or, at least, there need comparisons with different hyperparameters to claim consistent improvement.
>
> To find hyperparameters, we first tune them for optimal performance with PPO. For ITER, we then keep these PPO hyperparameters fixed and only tune t_distill (all other hyper-parameters were kept fixed after some preliminary experiments). Consequently, independent tuning of hyperparameters could therefore only improve PPO+ITER, not PPO.
> We chose not to independently tune PPO+ITER for computational reasons and to prevent overfitting of hyperparameters for our method.

---

> > ### Comment · AnonReviewer4 · 2020-11-24
> > **Still concern on the clarity on the generalization that the authors aim to improve**
> >
> > Thanks for the response, which clearly addressed my concern on the fairness in the comparison between PPO and PPO+ITER. Apparently, the authors indeed tried their best on the fair comparison.
> >
> > However, I still doubt that ITER always provides performance improvement in generalization. Even in the updated manuscript, the range of generalization that can be improved by ITER is not clearly defined. This is really important. Indeed, according to the recent works on analysis of deep learning, e.g., Gunasekar, Suriya, et al. "Implicit bias of gradient descent on linear convolutional networks." Advances in Neural Information Processing Systems. 2018., the impact of re-learning might depend on not only the neural network architecture but also optimization method. The authors need to investigate such dependencies of neural network architecture and update algorithm. However, currently, the authors just argue the mysterious "generalization effect" from re-learning. I'm not sure such re-learning will improve ANY kinds of generalization.
> >
> > Therefore, I don't think this work is ready to be published. In addition, the idea of re-learning is somewhat universal in the context of continual learning/life-long learning. I keep my original assessment.

---

> > > ### Author Response · Authors · 2020-11-24
> > > **-**
> > >
> > > While we cannot provide theoretical guarantees, we provide empirical evidence in multiple experiments that re-learning can improve generalisation in the situations discussed where transient non-stationarity negatively affected generalisation.
> > >
> > > The architectures and optimizers we use are standard in the community and cover multiple cases: our SL experiments use a (adapted) ResNet with SGD while our RL experiments use Adam with either a CNN architecture for the gridworlds or the widely used Impala architecture for ProcGen.
> > > Consequently, we expect our results to be applicable to many settings that are of relevance to the community.
> > >
> > > Lastly, as noted in our related work section, re-learning is used very differently in the context of continual or life-long learning: While in those cases one tries to prevent forgetting of early data, in our case forgetting the impact of earlier data is precisely our goal.

---

### Author Response · Authors · 2020-11-19
**Updated Paper**

Dear Reviewers,

thank you for your thoughtful reviews!

We have updated the paper to incorporate your feedback. Apart from smaller modifications, the two main changes are
* An additional experiment on a new environment (Sokoban) in section 5.2 to provide more RL results. (Some seeds are still running, explaining the sudden change in standard error at around 1.4e8)
* A more detailed explanation of the supervised learning experiments in section 5.4 (previously 5.3), including a new schematic figure (fig. 4) explaining the training setup.

We've also responded to your reviews individually and are looking forward to any further comments, questions or feedback.

---

### Decision · Program_Chairs · 2021-01-07
**Final Decision**

**Decision:**

Accept (Poster)

**Comment:**

There is a substantial contribution in identifying novel questions/issues, as this paper certainly does. Neither I nor the reviewers have seen this issue of transient non-stationary before, and the authors make a compelling case for it, especially in the supervised setting with the CIFAR experiments. It is less compelling through the RL experiments. As such, this paper is likely to inspire new work within the field. To me, Figure 1 is the most interesting aspect of the whole paper.

The initial approach by the authors is questionable in its effectiveness, and is likely to be improved by others in the future. Some of the results in Figure 3 are questionable, especially when you look at the individual curves in Figure 8. So overall, this means that the authors have identified a truly novel issue, and proposed an initial method that is just okay.  They've done a nice job investigating this in a supervised setting, and need to push further in the RL setting.

The question is whether the novel contribution of the problem outweighs that the algorithm and its evaluation could use improvement.  The reviewers debated this in the discussion, with points on both sides, but the novelty of the question/issue (even if the investigation could use work) is likely to inspire further work in this direction.

Other notes:
The authors could have evaluated the (impractical) version of their algorithm proposed in the first paragraph of Section 4.2. This would inform 1) whether their parallel training approximation is close to the optimal algorithm, and 2) whether the optimal (impractical) algorithm is capable of improving generalization significantly. If the latter is true, it would leave open a huge avenue of investigation to find better approximate solutions.